# Heterogeneous N₂O₅ reactions on atmospheric aerosols at four Chinese sites: Improving model representation of uptake parameters

Chuan Yu[1,2], Zhe Wang[3], Men Xia[2], Xiao Fu[2], Weihao Wang[2], Yee Jun Tham[2,4], Tianshu Chen[1], Penggang Zheng[1], Hongyong Li[1], Ye Shan[1], Xinfeng Wang[1], Likun Xue[1], Yan Zhou[5], Dingli Yue[5], Yubo Ou[5], Jian Gao[6], Keding Lu[7], Steven S. Brown[8,9], Yuanhang Zhang[7], Tao Wang[2]

[1]Environment Research Institute, Shandong University, Ji'nan, Shandong, China
[2]Department of Civil and Environmental Engineering, The Hong Kong Polytechnic University, Hong Kong, China
[3]Division of Environment and Sustainability, The Hong Kong University of Science and Technology.
[4]Institute for Atmospheric and Earth System Research/Physics, University of Helsinki, 00014, Helsinki, Finland
[5]Guangdong Environmental Monitoring Center, State Environmental Protection Key Laboratory of Regional Air Quality Monitoring, Guangzhou, China
[6]Chinese Research Academy of Environmental Sciences, Beijing, China
[7]State Key Joint Laboratory of Environmental Simulation and Pollution Control, College of Environmental Sciences and Engineering, Peking University, Beijing, China
[8]Chemical Sciences Division, Earth System Research Laboratory, NOAA, Boulder, CO, USA
[9]Department of Chemistry, University of Colorado Boulder, Boulder, CO, USA.

*Correspondence to*: Zhe Wang (z.wang@ust.hk) or Tao Wang (cetwang@polyu.edu.hk)

**Abstract.** Heterogeneous reactivity of N₂O₅ on aerosols is a critical parameter in assessing NOₓ fate, nitrate production, and particulate chloride activation. Accurate measurement of its uptake coefficient ($\gamma_{N2O5}$) and representation in air quality models are challenging, especially in the polluted environment. With an in-situ aerosol flow tube system, the $\gamma_{N2O5}$ was directly measured on ambient aerosols at two rural sites in northern and southern China. The results were analyzed together with the $\gamma_{N2O5}$ derived from previous field studies in China to obtain a holistic picture of N₂O₅ uptake and the influencing factors under various climatic and chemical conditions. The field derived/measured $\gamma_{N2O5}$ was generally promoted by the aerosol water content and suppressed by particle nitrate. Significant discrepancies were found between the measured $\gamma_{N2O5}$ and that estimated from laboratory-determined parameterizations. An observation-based empirical parameterization was derived in the present work, which better reproduced the mean value and variability of the observed $\gamma_{N2O5}$. Incorporating this new parameterization in a regional air quality model (WRF-CMAQ) has improved the simulation of nitrogen oxides and secondary nitrate in the polluted regions of China.

## 1 Introduction

Heterogeneous reaction of dinitrogen pentoxide (N₂O₅) on aerosol surfaces plays an important role in the nocturnal removal of nitrogen oxides (NOₓ), secondary nitrate formation, and chlorine activation through nitryl chloride (ClNO₂) production on chloride-containing aerosols (Brown et al., 2006; Osthoff et al., 2008; Thornton et al., 2010; Wang et al., 2016). Realistically representing this process in air quality models is therefore necessary for the prediction and mitigation of ground-level ozone

and particulate pollution. The currently accepted mechanism of the heterogeneous reaction of $N_2O_5$ on aqueous aerosols starts with the mass accommodation of $N_2O_5$ on aerosol surface (R1), followed by reversible $N_2O_5$ hydrolysis to form nitrate and intermediate $H_2ONO_2^+$ in the aqueous phase (R2). The intermediate $H_2ONO_2^+$ will react with $H_2O$ or $Cl^-$ to form $HNO_3$ or $ClNO_2$, respectively (R3 and R4) (Behnke et al., 1997; Finlayson-Pitts et al., 1989; Schweitzer et al., 1998; Thornton and Abbatt, 2005; Bertram and Thornton, 2009).

$$N_2O_5(g) \xrightarrow{k_1} N_2O_5(aq) \tag{R1}$$

$$N_2O_5(aq) + H_2O(l) \underset{k_{2b}}{\overset{k_{2f}}{\rightleftharpoons}} H_2ONO_2^+(aq) + NO_3^-(aq) \tag{R2}$$

$$H_2ONO_2^+(aq) + H_2O(l) \xrightarrow{k_3} H_3O^+(aq) + HNO_3(aq) \tag{R3}$$

$$H_2ONO_2^+(aq) + Cl^-(aq) \xrightarrow{k_4} ClNO_2(g) + H_2O(l) \tag{R4}$$

The reaction probability of $N_2O_5$, the so-called uptake coefficient $\gamma_{N2O5}$, is the fraction of $N_2O_5$ net removal upon collisions on aerosols, and is a key parameter to describe the heterogeneous loss rate of $N_2O_5$ on ambient aerosols. $\gamma_{N2O5}$ was first measured using aerosol flow tube in the laboratory and was shown dependent on aerosol chemical compositions such as water content, nitrate concentration, chloride concentration and organic coatings. Specifically, the aerosol water content can enhance the $N_2O_5$ uptake by promoting the hydrolysis of $N_2O_5$ (e.g. Hallquist et al., 2003; Thornton et al., 2003), while nitrate favors the reverse of reaction (R2) and thus suppress the $N_2O_5$ uptake (e.g. Wahner et al., 1998; Bertram & Thornton, 2009). On the contrary, chloride in the aqueous aerosol will react with the intermediate $H_2ONO_2^+$ faster than $NO_3^-$ and negate the nitrate suppression effect (e.g. Behnke et al., 1997; Bertram & Thornton, 2009). Organic coatings also can suppress $N_2O_5$ uptake by inhibiting the mass accommodation of $N_2O_5$ or limiting the availability of liquid water on the aerosol surface (e.g. Thornton & Abbatt, 2005; Anttila et al., 2006; Cosman et al., 2008; Gaston et al., 2014). Based on the laboratory studies, several parameterizations have been proposed to predict the variations of $\gamma_{N2O5}$, with considerations of temperature, RH, aerosol water content, nitrate, chloride, aerosol volume to surface area ratio and organic coatings (Davis et al., 2008; Evans & Jacob, 2005; Anttila et al., 2006; Riemer et al., 2009; Griffiths et al., 2009; Bertram & Thornton, 2009).

To investigate the heterogeneous process of $N_2O_5$ in ambient environments, $\gamma_{N2O5}$ was also derived from ambient concentrations of $N_2O_5$ with several methods, including steady-state lifetime estimation (Brown et al., 2006; Brown et al., 2009; Brown et al., 2016), secondary products formation rate determination (Phillips et al., 2016), and inverse iterative box model calculation (Wagner et al., 2013). In addition, aerosol flow tubes have been deployed to the field solely or in combination with an iterative model to directly 'measure' $\gamma_{N2O5}$ on ambient aerosols (Bertram et al., 2009a; Wang et al., 2018). Several studies have compared the field-derived/measured $\gamma_{N2O5}$ with that calculated from the parameterizations based on the laboratory results, which revealed significant discrepancies between them and large variations in the relationship between $\gamma_{N2O5}$ and aerosol chemical composition (e.g. Bertram et al., 2009b; Riedel et al., 2016; Morgan et al., 2015; Wang et al., 2017a; Tham

et al., 2018; McDuffie et al., 2018a). Recently, McDuffie et al. (2018a) proposed an empirical parameterization based on the aircraft measurements of $N_2O_5$ in the eastern United States, which can reproduce the mean value of the field-derived $\gamma_{N2O5}$ but still has difficulty in explaining its large variability. The discrepancies between the field-derived/measured and parameterized $\gamma_{N2O5}$ lie in the differences between the complex aerosols in ambient conditions and the simple proxies used in laboratory studies, for example, more complex organic matters or mixing state of ambient aerosols, and highlight the demand for the further comprehensive investigation of $N_2O_5$ uptake in diverse atmospheric conditions.

To further investigate the active $N_2O_5$ heterogeneous process revealed in previously studies in China (e.g., Wang et al., 2017a; Wang et al., 2017c; Tham et al., 2018; Yun et al., 2019), direct measurements of $\gamma_{N2O5}$ were conducted at two rural sites in northern and southern China in this work, by using the recently improved aerosol flow-tube system (Wang et al., 2018). Integrating them with the previous field results in various regions of China, we examine in detail the key factors that determine the $\gamma_{N2O5}$ and compare them with laboratory-derived parameterizations. Then we propose improved parameters for $\gamma_{N2O5}$ to better represent the $N_2O_5$ reactivity in polluted regions of China, and model simulations with incorporation of the new parameters were also performed to evaluate its representativeness and applicability.

## 2 Method

Field measurements of $\gamma_{N2O5}$ and related parameters were conducted at a semi-rural site (Heshan) in southern China from 22 February to 28 March 2017 and at a mountain site (Mt. Tai) in northern China from 11 March to 8 April 2018. Heshan site was located on a small hill (22.73°N, 112.92°E, 60 m a.s.l), surrounded by subtropical trees and some farmlands. A small city, Heshan, is 10 km to the northeast of the site, and three large cities, Guangzhou (the capital of Guangdong Province), Foshan and Jiangmen, are 80 km to the northeast, 50 km to the northeast and 30 km to the southwest of the site, respectively. The site is affected by vehicle emissions from three highways and two provincial roads within 10 km and some residential/agriculture activities in the area, and thus was considered as a semi-rural site. Mt. Tai site was located on the top of Mount Tai (36.25°N, 117.10°E, 1545 m a.s.l) in Shandong Province, and is affected by regional air pollution with limited impact from local sources. Two cities of Tai'an and Jinan (the capital of Shandong Province) are 15 km and 60 km to the south and north, respectively. $N_2O_5$ and $ClNO_2$ were measured using an iodide-adduct chemical ionization mass spectrometer (CIMS; THS Instrument, Atlanta), which has been deployed in several field campaigns (Wang et al., 2016; Tham et al., 2016; Wang et al., 2017a; Wang et al., 2017b; Yun et al., 2018). The related trace gases ($O_3$, $NO/NO_2$, etc.), aerosols size distribution, aerosol composition, and meteorological parameters were concurrently measured during the campaigns. Detailed descriptions of the measurement site and instrumentation can be found in Yun et al. (2018) and Wang et al. (2017a), and the measurement techniques, uncertainties and detection limits of the instruments are summarized in Table S1.

The uptake coefficient of $N_2O_5$, $\gamma_{N2O5}$, was derived from the direct measurement of the loss rate coefficient of $N_2O_5$ on ambient aerosols using an aerosol flow tube based on the design of Bertram et al. (2009), with some improvements and coupling with an iterative box model for polluted environments (Wang et al., 2018). Briefly, the flow tube consisted of a cylindrical stainless-

steel tube of 12.5 cm inner diameter and 120 cm length, with two 10 cm deep 60° tapered caps. The inner wall of the flow tube was coated with Teflon to reduce the wall loss of $N_2O_5$. The inlet was equipped with parallel sampling pass ways with one having a filter to remove aerosols. The switch of stainless steel valves allows the ambient air with or without aerosols to be introduced into the flow tube. The in-situ generated $N_2O_5$ (4.3 ppbv at 120 mL min$^{-1}$, produced from the reaction of $O_3$ with excess $NO_2$) was added to the ambient air after the valves and prior to the flow tube by a side port. The total flow rate in the flow tube was 4.6 L min$^{-1}$, corresponding to a residence time of 149 s. During the flow tube experiments, the $N_2O_5$, NO, $NO_2$, $O_3$, particle number and size distribution, and RH were simultaneously measured at the base of the flow tube, and ambient NO, $NO_2$, and $O_3$ were also measured at the same time.

An iterative box model considering multiple reactions of production and loss of $N_2O_5$ (Reactions R5–R10) was used to determine the loss rate of $N_2O_5$ in both aerosol and non-aerosol modes (Wang et. al, 2018).

$$O_3 + NO_2 \rightarrow NO_3 + O_2 \tag{R5}$$

$$NO_3 + NO_2 \rightarrow N_2O_5 \tag{R6}$$

$$O_3 + NO \rightarrow NO_2 + O_2 \tag{R7}$$

$$NO_3 + NO \rightarrow 2NO_2 \tag{R8}$$

$$NO_3 + VOC \rightarrow products \tag{R9}$$

$$N_2O_5 + aerosol/wall \rightarrow products \tag{R10}$$

The rate constants of reactions (R5) to (R8) were adopted from Sander et al. (2009), and that of reaction (R9) was from Atkinson and Arey (2003). With the constraint of measurement data at the entrance of the flow tube reactor in the model, the exit concentrations of $NO_2$, $O_3$, and $N_2O_5$ can be predicted by integrating these reactions. The $N_2O_5$ loss rate coefficient, $k_{10}$, was adjusted until the $N_2O_5$ concentration predicted by the iterative box model matched with the measured $N_2O_5$ value at the exit. Then the loss rate coefficient of $N_2O_5$ on aerosols surfaces can be determined from the differences of $k_{10}$ with or without aerosol, assuming a constant $k_{wall}$ in both modes. The uptake coefficient of $N_2O_5$ on ambient aerosol is then calculated by:

$$\gamma_{N2O5} = (k_{10}^{w/aerosol} - k_{10}^{wo/aerosol})/(cS_a). \tag{Eq. (1)}$$

The $k_{10}^{w/aerosol}$ and $k_{10}^{wo/aerosol}$ are the $N_2O_5$ loss rate coefficient with or without aerosol, and $c$ is the mean molecular speed of $N_2O_5$, and $S_a$ is the particle surface area. The ambient aerosol surface area density was calculated from the dry particle size distributions corrected with a size-resolved kappa-Köhler function and ambient RH (Hennig et al., 2005; Liu et al., 2014; Yun et al., 2018). By assuming an uncertainty of 20% in the particle number size distribution introduced by charging efficiency, sizing accuracy and flow rate variability (Jiang et al., 2014; Kuang et al., 2016; Widensohler et al. 2014) and an uncertainty of 15% for the hygroscopic growth at RH<90% (Liu et al., 2014), the uncertainty associated with $S_a$ measurement was estimated to be approximately 30%. It has to be noted that the uncertainty introduced by the particle morphology was not accounted for here, and thus the reported uncertainty in $S_a$ can be considered as a lower limit. In addition, Wang et al. (2018) employed a

Monte Carlo approach to evaluate the uncertainty in $\gamma_{N2O5}$ determination from different parameters in the flow tube system, including mean residence time, wall loss variability with ambient RH, input $N_2O_5$ concentration, as well as the variability of ambient NO, $NO_2$, $O_3$, and VOCs levels during a measurement cycle. The estimated overall uncertainty in $\gamma_{N2O5}$ determination was propagated to be 37% to 40% at $\gamma_{N2O5}$ around 0.03 and from 34% to 65% at $\gamma_{N2O5}$ around 0.01 when RH varied from 20% to 70% (Wang et al., 2018). The uncertainty would be increased for higher RH conditions, even up to 100% for RH ≥90% (Wang et al., 2017).

To obtain a holistic picture of the $\gamma_{N2O5}$ in different geographic regions of China, field measurement results from three previous campaigns are also used in the present study. These measurements were conducted at a sub-rural site at Wangdu and the same mountain site at Mt. Tai in 2014, and a mountain site at Mt. Tai Mo Shan in South China in 2016. All the sites are regionally representative sites, as they are situated in an area with limited anthropogenic influences (Tham et al., 2016; Wang et al., 2017a; Yun et al., 2018; Wang et al., 2016). The detailed information of the sampling sites, instrumentation and $\gamma_{N2O5}$ determination approach have been described in the previous publications (Wang et al., 2016; Tham et al., 2016; Wang et al., 2017a), and site descriptions are briefly summarized in the SI. The locations of all the measurement sites are shown on the map in Fig. 1a. The statistics of the trace gases and $PM_{2.5}$ measured during the campaigns were summarized in Fig. 1b, representing general pollution conditions at these sites. The mean concentration of $O_3$, NOx and $PM_{2.5}$ at these sites ranged from 43 to 80 ppbv, 2.4 to 14.5 ppbv and 9.9 to 80.2 μg m$^{-3}$, respectively.

In addition, the Community Multiscale Air Quality (CMAQ) model (v5.1) was employed to evaluate the uptake parameterization. Two simulations (default and revised) were conducted. In the default case, the $N_2O_5$ uptake and $ClNO_2$ production were calculated based on the parameterization of Bertram and Thornton (2009). In the revised case, the new parameterization derived in this study was used. Other model configurations were the same. The SAPRC07tic gas mechanism and AERO6i aerosol mechanism was used. Weather Research and Forecasting (WRF) (v4.0) was applied to generate the meteorological inputs for the CMAQ simulations. The anthropogenic emission inputs were generated based on the local Chinese emission inventory (Zhao et al. 2018) and the INTEX-B dataset for Asia (Zhang et al., 2009). The high-resolution chloride emission inventory for China from Fu et al. (2018) was also included. More details for model configuration can be found in Fu et al. (2019). The simulation domain covers China with a resolution of 36×36 km (Fig. S1), based on a Lambert projection with two true latitudes of 25°N and 40°N. The simulation period was from 1 to 31 December 2017, with five days before as a spin-up time.

## 3 Results and discussion

### 3.1 Field measured $\gamma_{N2O5}$ and influencing factors

During the field measurements at Heshan and Mt. Tai, the air was characterized as moderately polluted for $O_3$ (43±22 ppbv at Heshan and 63±14 ppbv at Mt. Tai), $NO_x$ (14.0±11.5 ppbv at Heshan and 2.2±2.1 ppbv at Mt. Tai), and $PM_{2.5}$ (66.7±41.9 μg m$^{-3}$ at Heshan and 33.7±26.7 μg m$^{-3}$ at Mt. Tai), as summarized in Table S2 and shown in Fig 1b. $\gamma_{N2O5}$, which was directly

measured using the aerosol flow tube, showed large variation ranging from 0.002 to 0.067 with an average of 0.020±0.019 at

Heshan, and from 0.001 to 0.019 with an average of 0.011±0.005 at Mt. Tai. These values are within the range of $10^{-5}$ to > 0.1 derived from the ambient $N_2O_5$ concentrations around the world (e.g. Brow et al., 2006; Bertram et al., 2009b; Riedel et al., 2016; Morgan et al., 2015; Wang et al., 2017a; Tham et al., 2018; McDuffie et al., 2018a), but slightly lower than the previous results in the polluted regions in China (0.021 to 0.102) (Wang et al., 2017a; Wang et al., 2017b; Wang et al., 2017c). The field measured $\gamma_{N2O5}$ and relevant pollutants at the two sites and those derived from three previous studies in China are

summarized in Fig 1b, covering diverse atmospheric conditions from moderately humid to humid conditions and from clean to polluted conditions.

Figure 2 shows the relationship of the field measured $\gamma_{N2O5}$ with the aerosol composition during five campaigns at those four sites in China. It can be seen that the $\gamma_{N2O5}$ had a good positive correlation with the aerosol water content ($r^2 = 0.65$) (Fig. 2a), suggesting a common controlling role of aerosol water in the reactivity of $N_2O_5$ in both northern and southern China. Although

the positive correlation of $\gamma_{N2O5}$ with the humidity or aerosol water has been observed in the low range in previous laboratory studies, the $\gamma_{N2O5}$ reached plateaus at a value around 0.036 at RH> 50% or [$H_2O$] > 15 M (Hallquist et al., 2003; Thornton et al., 2003; Bertram & Thornton, 2009). In contrast, other laboratory studies also measured higher $\gamma_{N2O5}$ values on $NH_4HSO_4$ particles. For example, Mozurkewich and Calvert (1988) reported an upper limit of $\gamma_{N2O5}$ of 0.056 at RH = 55% at 293 K, which increased to around 0.1 at 274 K. Kane et al. (2001) observed a strong RH dependent $\gamma_{N2O5,}$ increasing from 0.018 to

0.069 with RH from 56% to 99%, which is largely consistent with the field results in the present study. Moreover, several field measurements also observed $\gamma_{N2O5}$ value exceeding 0.04 at high RH or water molarity (e.g. Philips et al., 2016; McDuffie et. al, 2018a; Wang et al., 2017c; Tham et. al, 2018), and some of them also found the similar positive relationship between $\gamma_{N2O5}$ and water molarity (McDuffie et. al, 2018a). Although uncertainties may exist in the calculation of aerosol surface and uptake coefficient at high ambient RH conditions, our results with a consistently increasing trend of $\gamma_{N2O5}$ with [$H_2O$] from below 10

M up to 50 M suggest that the aerosol water content strongly affects the activity of $N_2O_5$ uptake, and that $N_2O_5$ hydrolysis is always limited by aerosol water content under all the encountered ambient conditions. Since limited measurement data of $\gamma_{N2O5}$ from laboratory and fields are available at RH > 80% condition, it is unclear what exact mechanism or process (e.g., phase change different from laboratory-made particles or acidity involved) promote more effective uptake on ambient aerosols at higher aerosol water content condition. Therefore, more detailed investigations of $N_2O_5$ uptake on nano-size water/aerosol

droplets in the real (or close to real) ambient conditions are clearly warranted. For nitrate, a clear suppression effect can be found at the Chinese sites (Fig. 2b), which is similar to most of the previous field and laboratory studies. The decrease of $\gamma_{N2O5}$ with increasing nitrate concentration seems to be better captured by an 'exponential-decay' curve, with almost linear suppression for [$NO_3^-$] below 5 M. The observed $\gamma_{N2O5}$ under high nitrate condition (> 5 M) was generally below 0.025 and became nitrate independent as the nitrate levels further increased.

The $\gamma_{N2O5}$ variation was affected by the additive effects from both [$NO_3^-$] and [$H_2O$], which could not be easily isolated because of their competition reactions with the reactive intermediate $H_2ONO_2^+$. This is further supported by the positive dependence of $\gamma_{N2O5}$ on the molar ratio of [$H_2O$]/[$NO_3^-$] (Fig. 2c). Different from the previously reported plateauing of $\gamma_{N2O5}$ with increasing

[H$_2$O]/[NO$_3^-$] ratio in laboratory studies (Hallquist et al., 2003; Bertram & Thornton, 2009), no decrease in γ$_{N2O5}$ suppression was found in the present study for [H$_2$O]/[NO$_3^-$] ratio of up to 60. The more scattered data at higher [H$_2$O]/[NO$_3^-$] range implies
that the variation of γ$_{N2O5}$ become more sensitive to other factors in the diluted aqueous aerosols. Although the γ$_{N2O5}$ measured at two mountain sites showed a positive relationship with [Cl$^-$]/[NO$_3^-$], the overall results from five sites did not exhibit an obvious pattern (Fig. 2d). These results suggest that chloride concentration may not play a critical role in γ$_{N2O5}$ during our observations as laboratory studies have observed (Bertram & Thornton, 2009), possibly due to the complex effect of aerosol mixing state. Though the measured γ$_{N2O5}$ exhibited nonlinear relationship and complex dependence on different factors at a
single site, the general consistent patterns at different sites in this study suggests the feasibility of a common parameterization representing the N$_2$O$_5$ uptake in these regions.

## 3.2 Comparison to parameterizations

Current regional air quality models such as WRF-Chem and CMAQ mainly use the γ$_{N2O5}$ parameterization recommended by Bertram and Thornton (2009) (hereafter referred to BT09), which links γ$_{N2O5}$ to aerosol water content, nitrate and chloride as
well as the aerosol size and ambient temperature. The BT09 parameterization based on the above-mentioned reaction mechanism (R1-R5) was expressed as follows:

$$\gamma_{N2O5} = \frac{4}{c}\frac{V_a}{S_a}K_H k'_{2f}\left(1 - \frac{1}{\left(\frac{k_3[H_2O]}{k_{2b}[NO_3^-]}\right) + 1 + \left(\frac{k_4[Cl^-]}{k_{2b}[NO_3^-]}\right)}\right)$$  Eq. (2)

$$k'_{2f} = \beta - \beta e^{(-\delta[H_2O])}$$  Eq. (3)

where $V_a/S_a$ is the measured aerosol volume to surface area ratio, ranging from 3.30×10$^{-8}$ to 9.29×10$^{-8}$ m in the five campaigns;
$K_H$ is Henry's law coefficient, taken as 51 (Bertram & Thornton, 2009; Fried et al., 1994); $\beta$ = 1.15×10$^6$; $\delta$ = -0.13. k$_3$/k$_{2b}$ (= 0.06) and k$_4$/k$_{2b}$ (= 29) represent the relative rates of competing reactions of intermediate H$_2$ONO$_2^+$(aq) with H$_2$O (R3) and Cl$^-$ (R4) over NO$_3^-$ (R2), respectively. [H$_2$O], [NO$_3^-$] and [Cl$^-$] are the aerosol water content, aerosol nitrate and chloride molarity, respectively, calculated by the E-AIM model with the measured ionic compositions of PM$_{2.5}$ and RH (http://www.aim.env.uea.ac.uk/aim/aim.php) (Wexler and Clegg, 2002).
We calculate the γ$_{N2O5}$ values from BT09 with the measured aerosol composition at the five sites. The parameterized γ$_{N2O5}$ ranged from 0.021 to 0.075, with an average of 0.047±0.015, which overestimates the observed values by a factor of 1.8. When the chloride effect was excluded, the parameterized γ$_{N2O5}$ mean value decreased to 0.020±0.018, which was better correlated with but underestimated (by 30%) the measurements (Table 1). Figure 3 compares the observation-derived and parameterized γ$_{N2O5}$ at five sites in China and in different parts of the world. The BT09 parameterization (blue markers in Fig. 3) generally
overestimates the observed γ$_{N2O5}$ values in the range of 0.001 to 0.03, but is closer (within a factor of 1.5) to the observed value for γ$_{N2O5}$ above 0.03 in Germany (Phillips et al., 2016) and Mt. Tai (Wang et al., 2017a). The BT09 parameterization excluding chloride effects (yellow markers) gives much better agreement, with more values located in the range within a factor of 2,

though the $\gamma_{N2O5}$ was still overpredicted in most of the studies in North America (Bertram et al., 2009b; Riedel et al.,2012; McDuffie et al., 2018a) except for Boulder (Wagner et al., 2013). The improvement indicates that the efficiency of chloride in competing for the $H_2ONO_2^+$ intermediate and the effects on $N_2O_5$ uptake on ambient aerosols might be overestimated, possibly due to the existence of other nucleophiles competing with $Cl^-$ (McDuffie et al., 2018b; Staudt et al., 2019), or different mixing states of particle and non-uniform distribution of available chlorine in the aerosols.

Organic matter/coating on the aerosols can suppress the uptake of $N_2O_5$ (Thornton & Abbatt, 2005; McNeill et al., 2006; Park et al., 2007), and previous studies have attempted to account for this effect by treating organics as a coating on the inorganic core (Anttila et al., 2006; Riemer et al., 2009). However, significant underpredictions were found from the parameterization of BT09 combined with the organic effect (Morgan et al., 2015; McDuffie et al., 2018a; Tham et al., 2018) (green and purple markers in Fig. 3). One reason could be that the parameterization does not differentiate the water-soluble organic fractions and simplifies the morphology and phase state, which leads to the underestimation of the solubility and/or diffusivity of $N_2O_5$ in the organics. The complex effects of organic matter on $N_2O_5$ uptake remain poorly quantified (McDuffie et al., 2018a), and the prediction of composition, morphology and phase state of the organic fractions are still difficult in current air quality models. Therefore, we do not consider the organic effect in deriving a new parameterization in the next section.

### 3.3 Observation-based empirical parameterization of $\gamma_{N2O5}$

Based on the above discussion and comparison, we attempt to derive a new empirical parameterization of $\gamma_{N2O5}$ following the BT09 framework (Eq. (2)) and using the measurement data from five field campaigns in China. The variables in the parameterization (i.e., reaction rates) were fitted with multiple regression to obtain the best representation of observations in China. The derived empirical parameterization of $\gamma_{N2O5}$ is shown as Eq. (4) and the fitted $\gamma_{N2O5}$ are summarized in Table 1.

$$\gamma_{N2O5} = \frac{4}{c}\frac{V_a}{S_a}K_H \times 3.0 \times 10^4 \times [H_2O]\left(1 - \frac{1}{\left(0.033 \times \frac{[H_2O]}{[NO_3^-]}\right) + 1 + \left(3.4 \times \frac{[Cl^-]}{[NO_3^-]}\right)}\right) \qquad \text{Eq. (4)}$$

In view of the linear dependence of $\gamma_{N2O5}$ on the aerosol water content in this study and reaction mechanism (Bertram & Thornton, 2009), the second-order reaction rate coefficient with water (refer to k'$_{2f}$ in Eq. (2) and Eq. (3)) was fitted as a linear function of [H$_2$O], as $(3.0 \pm 0.4) \times 10^4 \times [H_2O]$. This value is in reasonable agreement with the values of $(2.7-3.8) \times 10^4$, $\sim 3.9 \times 10^4$, and $2.6 \times 10^4$ M$^{-1}$ s$^{-1}$ determined from ammonium bisulfate, ammonium sulfate (Gaston et al., 2016) and aqueous organic acid particles (Thornton et al. 2003), respectively. Compared to original BT09 (Eq. (3)), the newly fitted k'$_{2f}$ is smaller for [H$_2$O] < 38 M, but become higher with the increasing of aerosol water content (Fig. S2). Different dimensionless K$_H$ values have been used in previous studies, e.g., $\sim$50 (e.g., Hallquist et al., 2003; Bertram and Thornton, 2009) or $\sim$120 (e.g., Gaston et al., 2014; 2016; Griffiths et al., 2009), which correspond to a Henry's law constant of 2 or 5 M atm$^{-1}$ at 298K. As $\gamma_{N2O5}$ in the parameterization is linearly dependent on the K$_H$, an increase of K$_H$ value would proportionally increase the $\gamma_{N2O5}$ value but cannot account for the large variability of measured $\gamma_{N2O5}$ values. Given the lack of an explicit function of effective Henry's

law constant for $N_2O_5$ to include the different process (e.g., 'salting-in' effect and surface processes), we use the value of 51 suggested by Bertram and Thornton (2009) and enclose those effects from the aerosol composition in the last 'chemical' term. The derived empirical ratios in the last 'chemical' term not only represent the competing ratio of these reactions but also include other unspecified effects or processes (e.g., organic coating, mixing state, other nucleophiles reactions, etc.). The fitted relative rates of competing reactions, i.e., $k_3/k_{2b}$ and $k_4/k_{2b}$, were $0.033\pm0.017$ and $3.4\pm1.4$, respectively, which are smaller than the original BT09 parameters by a factor of 1.8 and 8.5, respectively. The smaller ratios of the reaction rates indicate a smaller enhancement effect of chloride or a larger suppression effect by nitrate, which is consistent with the above-observed relationship of $\gamma_{N2O5}$ with the aerosol composition. Other suppression effects such as organic coating and mixing state that was not specified in the parameterization also may contribute to, and are reflected in the smaller fitted values. As compared in Fig. 4 and Table 1, the new empirical parameterization can better reproduce the average values and the large variability of the observed $\gamma_{N2O5}$ than the original BT09 both with and without considering $Cl^-$ effects.

As suggested by the previous studies, the production yield of $ClNO_2$ ($\Phi_{ClNO2}$) from $N_2O_5$ uptake is also a function of competing reactions of $H_2O$ and $Cl^-$ content in aerosols, and can be estimated from $\Phi_{ClNO2} = 1/(1+k_3/k_4\times[H_2O]/[Cl^-])$ (Bertram & Thornton, 2009). Based on the above-fitted results for $\gamma_{N2O5}$, $k_4/k_3$ is determined to be $105\pm37$ for the five sites, which is smaller than the values of $450\pm100$ (Roberts et al., 2009), $483\pm175$ (Bertram & Thornton, 2009) and $836\pm32$ (Behnke et al., 1997) derived from laboratory experiments and used in previous parameterizations. As compared in Fig. S3 and Table 1, although the newly fitted values improve the estimated $ClNO_2$ yield comparing to the original BT09 (with $k_4/k_3$ of 483), overestimation remains, and the large variability of observed $\Phi_{ClNO2}$ in different campaigns still cannot be well captured. As shown in Fig. 5, the new fits can better catch the $\Phi_{ClNO2}$ trend at $[H_2O]/[Cl^-] > 750$, but discrepancy is still obvious at $[H_2O]/[Cl^-] < 750$. The discrepancy could be due to aqueous-phase competition reactions of intermediate $H_2ONO_2^+$ with other compounds (e.g., phenol) (Heal et al., 2007), and $ClNO_2$ loss/reaction mechanisms (e.g., reaction with $Cl^-$ to form $Cl_2$) (Roberts et al., 2008, 2009). A recent laboratory study (Staudt et al. 2019) has reported that sulfate and acetate can suppress $\Phi_{ClNO2}$ for $Cl^-$ containing solutions, but such sulfate suppression effect was not observed in our results. Further studies are needed to identify and quantify these effects for better parameterizing the heterogeneous $ClNO_2$ production. Nonetheless, the revised $k_3/k_4$ from fitting the field data has improved the estimates of $\Phi_{ClNO2}$ at our study sites.

### 3.4 Evaluation of the empirical parameterization

To further evaluate the representativeness and validity of the newly fitted empirical parameterization of $\gamma_{N2O5}$ in predicting $N_2O_5$ heterogeneous process in air quality models, simulation tests were performed with the WRF-CMAQ model. The simulations were conducted with the incorporation of newly fitted and original BT09 parameterizations, respectively. The simulated concentrations of $NO_2$ and $NO_3^-$, as the key precursor and a product of the $N_2O_5$ uptake, were compared with the observed daily $NO_3^-$ concentrations at 28 sites and hourly $NO_2$ concentrations at 472 sites in the North China Plain during December of 2017. As summarized in Table 2 and shown in Fig. S4, the simulation with original BT09 parameterization overestimated the regionally averaged $NO_3^-$ concentrations by 18.7% compared to the observations, whereas the new

parameterization gave more consistent results with the observations ($20.98 \pm 18.77$ μg m$^{-3}$ vs $20.94 \pm 17.16$ μg m$^{-3}$), reducing the normalized mean bias (NMB) of simulated $NO_3^-$ concentration from 18.72% to 0.19 %. The simulated $NO_2$ concentrations were also in better agreement with the observations, with the NMB changed from -12.25 % to -8.06 %. In addition to $NO_2$ and $NO_3^-$, we also compared the simulated $N_2O_5$ concentrations for December 2017 with those observed in the wintertime at various locations of China, including two in the North China Plain (Beijing and Wangdu in Herbei province) and two in southern China (Tai Mao Shan and Heshan). As shown in Figure 6, with the new parameterization, the WRF-CMAQ model can better simulate the average concentration and variation range of $N_2O_5$ at these locations. Overall, the new parameterization has significantly reduced the discrepancies between the modeled and observed concentrations of $NO_2$, $N_2O_5$ and $NO_3^-$ at our study sites and periods in both northern and southern China. More tests of this empirical parameterization are warranted for other locations/seasons in China and other parts of the world.

## 4 Conclusion

Nitrate is becoming the predominant component of $PM_{2.5}$ during severe haze events in China in recent years (Zhang et al., 2015; Li et al., 2018), and ground-level ozone pollution in urban areas is also worsening (Wang et al., 2017d). Despite extensive research, current air quality models still have difficulties in accurately simulating the $N_2O_5$ uptake on aerosols, which limits their ability in predicting the lifetime and fate of $NO_x$ and therefore the productions of aerosol nitrate and ozone. Based on the measurements from five field campaigns at four sites across China with different atmospheric conditions, our study examined the factors influencing $N_2O_5$ uptake processes and derived an observation-based empirical parameterization of $N_2O_5$ uptake. While further research is still needed on the additional factors affecting $\gamma_{N2O5}$ and $\Phi_{ClNO2}$, the empirical parameterization derived here can be used in air quality models to improve the prediction of $PM_{2.5}$ and photochemical pollution in China and similar polluted regions of the world.

## Author contributions

TW and ZW designed the study. WW, CY and ZW designed the aerosol flow tube and CY carried out the aerosol flow tube measurements. MX, TC, PZ, HL, YS, YZ, and DY conducted the field measurement of relevant species and data analysis. XF performed the model simulation. CY, ZW, and TW wrote the manuscript, with discussions and comments from all co-authors.

## Competing interests.

The authors declare that they have no conflict of interest.

## Data availability

The data used in this study are available upon request from Zhe Wang (z.wang@ust.hk) and Tao Wang (cetwang@polyu.edu.hk).

315     ## Acknowledgments

This study is supported by grants from the National Natural Science Foundation of China (91544213, 91844301) and the Research Grants Council of Hong Kong Special Administrative Region, China (T24/504/17, C5022-14G, 15265516).

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

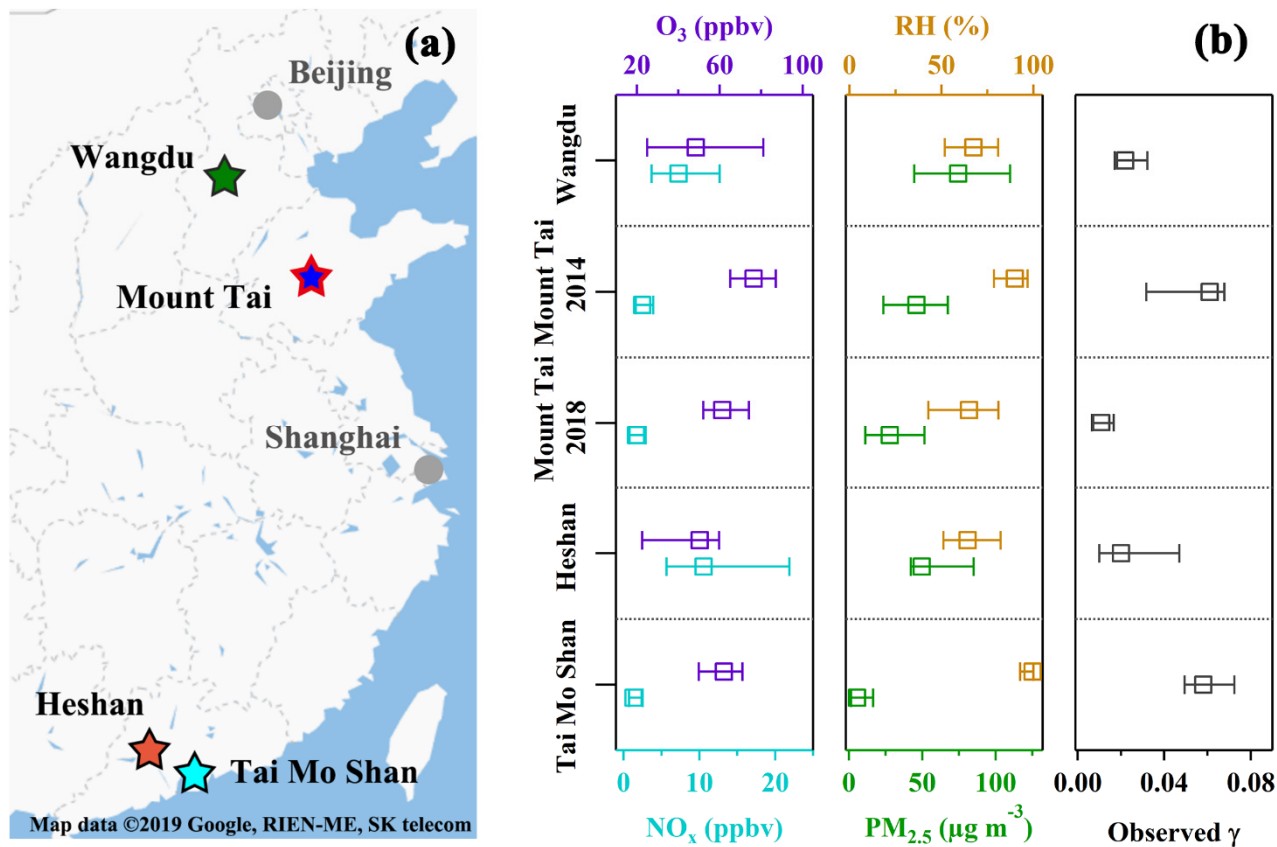


Figure 1. (a) The locations of the four field measurement sites (pentagram markers) in China. (b) Comparisons of the concentrations of $O_3$, $NO_x$, $PM_{2.5}$, and observed RH and $\gamma_{N2O5}$ during the five campaigns in China. Squares represent the median values and bars represent the interquartile ranges of the values in the five measurements. It should be noted that the high RH in Mt. Tai 2014 and Tai Mo Shan campaigns were caused by frequent cloud/fog events, and the $\gamma_{N2O5}$ was determined only during non-cloudy periods in these two campaigns.



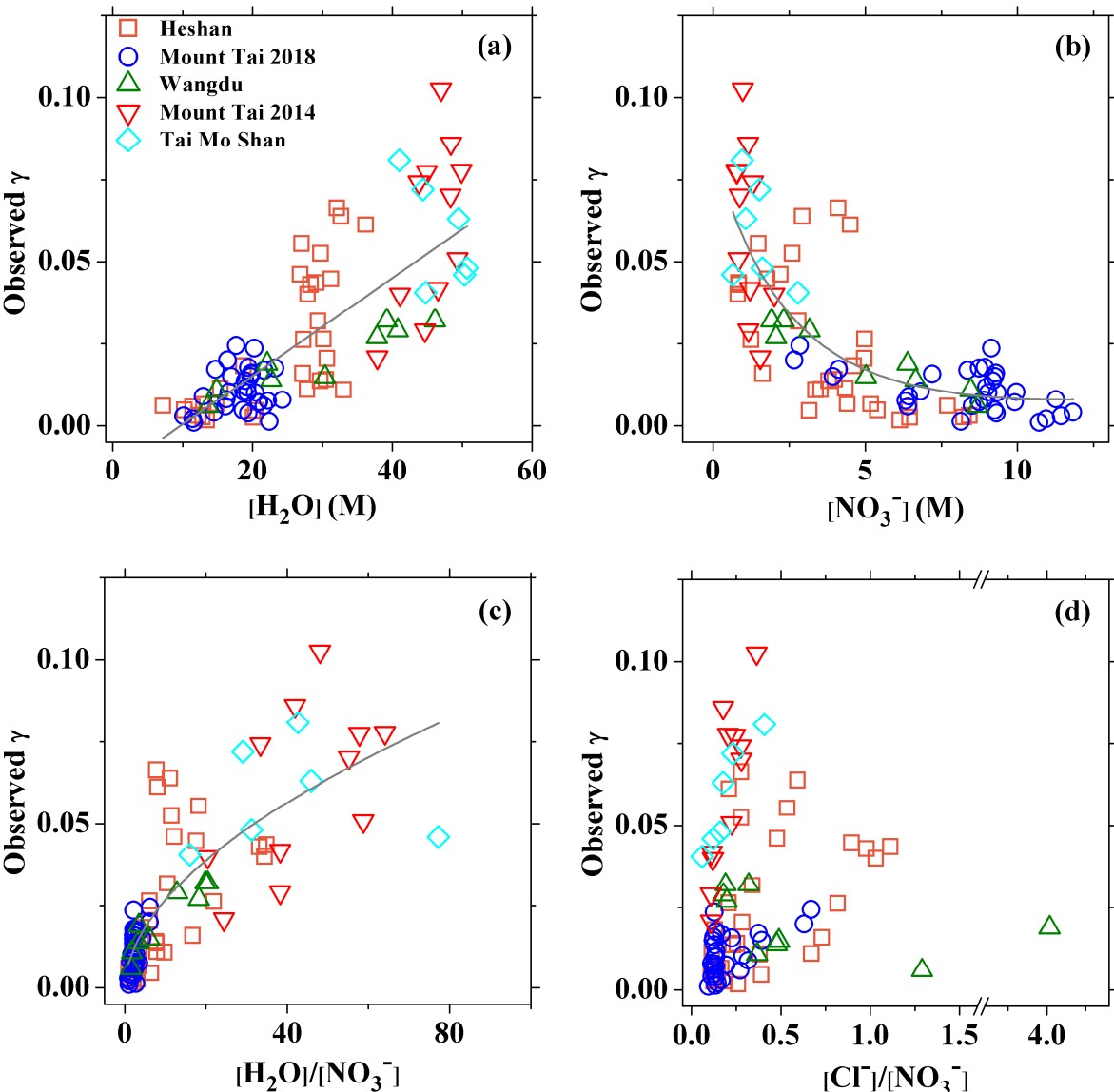


**Figure 2.** Relationship between the field measured/derived $N_2O_5$ uptake coefficient $\gamma_{N2O5}$ and (a) aerosol water content, (b) particle nitrate, (c) $H_2O$ to $NO_3^-$ molarity ratio, and (d) $Cl^-$ to $NO_3^-$ molarity ratio. Green triangles, red triangles, cyan squares, yellow squares and blue circles represent the results of Wangdu in 2014, Mount Tai in 2014, Tai Mo Shan in 2016, Heshan in 2017 and Mount Tai in 2018, respectively. The solid lines are linear or exponential regressions.

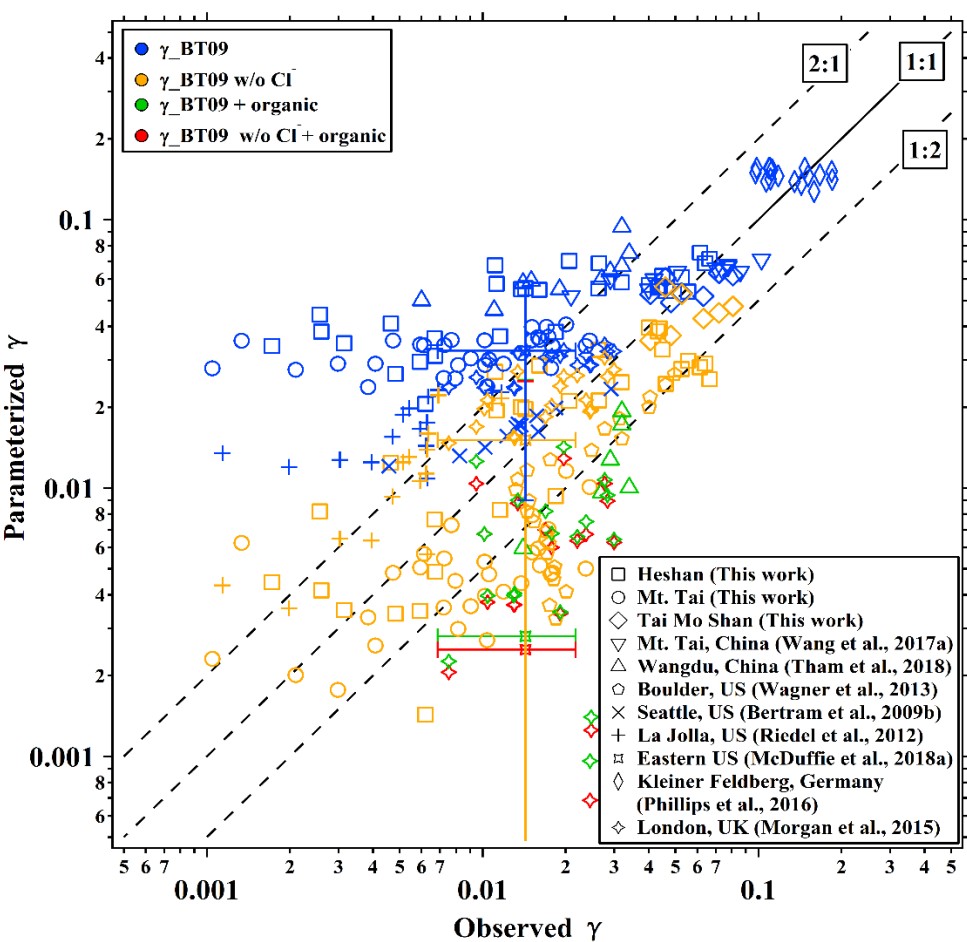


**Figure 3. Summary of the comparisons of field measured/derived $\gamma_{N2O5}$ and values estimated from parameterizations from the literature. Blue, yellow, green and purple markers represent the results calculated from parameterizations of original BT09, BT09 excluding chloride effect, BT09 plus organic effect, and BT09 excluding chloride but with organic effect, respectively.**


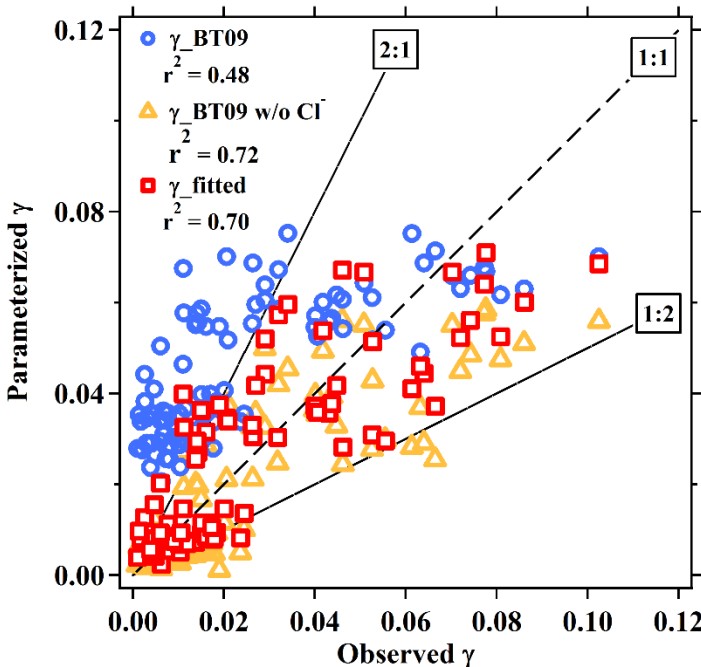

**Figure 4. Comparison of the field measured/derived $\gamma_{N2O5}$ with the values estimated from parameterizations for the five sites in the present study. The dashed line represents the 1:1 line. Blue circles, yellow triangles, and red squares are results estimated by BT09 parameterization, BT09 excluding chloride effect and the derived observation-based empirical parameterization, respectively.**

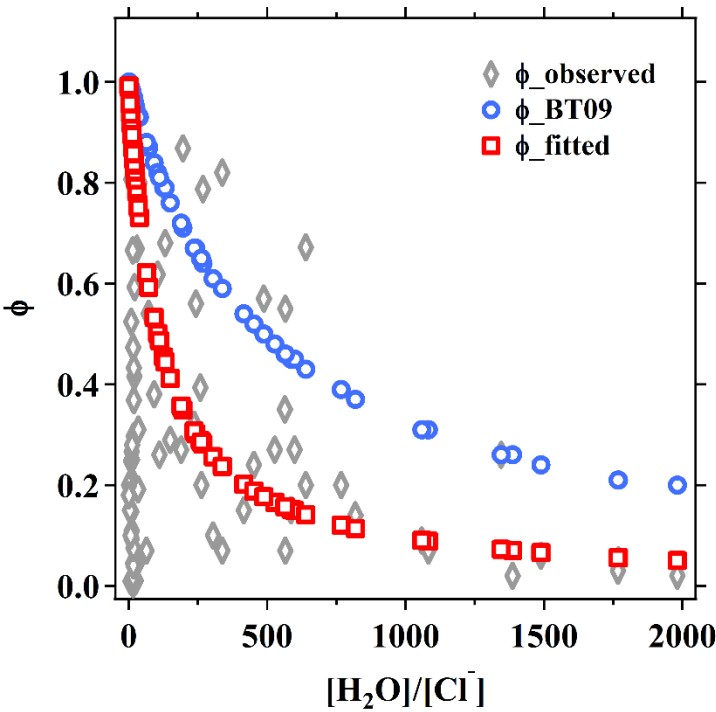

**Figure 5. Relationship between the ClNO$_2$ yield, $\Phi_{ClNO2}$, and the molarity ratio of H$_2$O to Cl$^-$. Grey rhombi, blue circles, and red squares represent the observed $\Phi_{ClNO2}$, values from BT09 parameterization and fitted from the empirical parameters derived in the present study, respectively.**

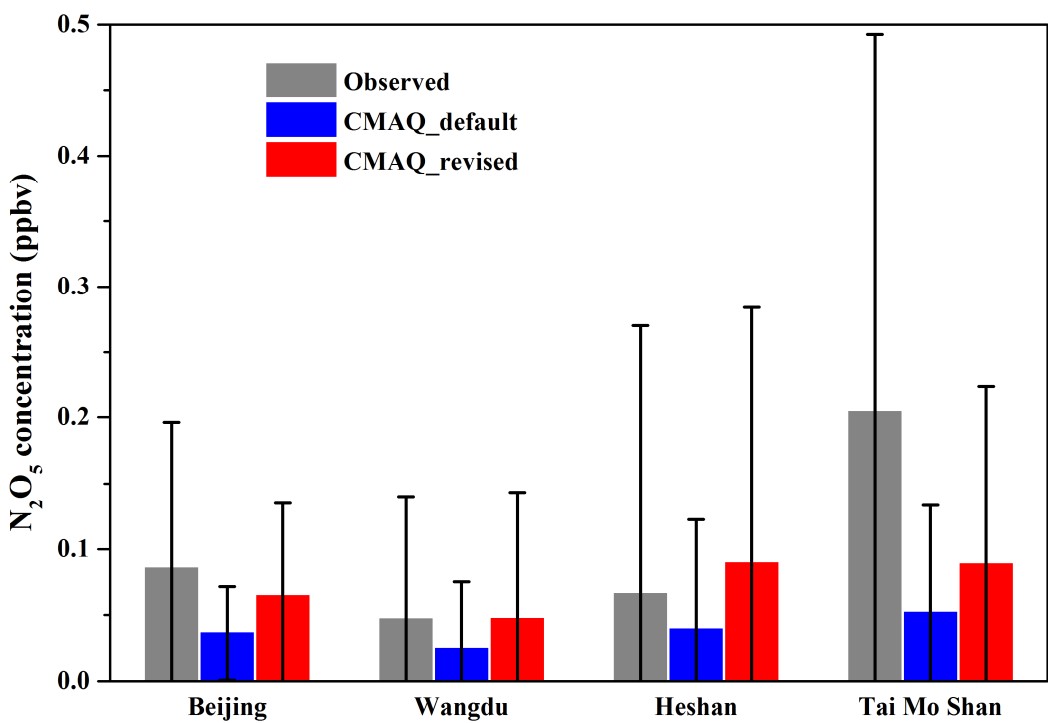

**Figure 6. Comparison of simulated $N_2O_5$ concentrations by the CMAQ model for December 2017 with the wintertime observation results from four sites in China. The field observations were conducted in Wangdu (Hebei Province) in December 2017, Beijing in January 2018, Heshan (Guangdong Province) in January 2017 and Tai Mao Shan (Hong Kong) in November 2013. The columns and error bars represent the average value and standard deviation, respectively.**





**Table 1. Statistical summary and comparison of the observed parameters (N$_2$O$_5$ uptake coefficient, $\gamma_{N2O5}$; ClNO$_2$ yield, $\Phi_{ClNO2}$) with values predicted from different parameterizations.**

| Parameters | | Average $\pm$ SD | Maximum | Minimum | $r^2$ |
|---|---|---|---|---|---|
| $\gamma_{N2O5}$ | Observed | 0.026 $\pm$ 0.024 | 0.10 | 0.001 | - |
| | BT09 | 0.047 $\pm$ 0.015 | 0.075 | 0.021 | 0.54 |
| | BT09 w/o Cl$^-$ | 0.020 $\pm$ 0.018 | 0.058 | 0.001 | 0.72 |
| | Fitted | 0.026 $\pm$ 0.020 | 0.071 | 0.002 | 0.70 |
| $\Phi_{ClNO2}$ | Observed | 0.31 $\pm$ 0.27 | 1.04 | 0.004 | - |
| | BT09 | 0.74 $\pm$ 0.26 | 1.00 | 0.20 | 0.025 |
| | Fitted | 0.57 $\pm$ 0.33 | 0.99 | 0.05 | 0.003 |





**Table 2. Statistical summary and comparison of the observed species (nitrate and NO$_2$ concentrations) with values predicted from different parameterization. NMB represents the normalized mean bias.**

| Species | | Simulated average ± SD (µg m$^{-3}$) | Observed average ± SD (µg m$^{-3}$) | NMB (%) | r$^2$ |
|---|---|---|---|---|---|
| NO$_3^-$ | CMAQ_default | 24.86 ± 20.48 | 20.94 ± 17.16 | 18.72 | 0.56 |
| | CMAQ_revised | 20.98 ± 18.77 | | 0.19 | 0.56 |
| NO$_2$ | CMAQ_default | 45.71 ± 31.21 | 52.09 ± 27.25 | -12.25 | 0.31 |
| | CMAQ_revised | 47.89 ± 32.10 | | -8.06 | 0.34 |
