# Peer review of "Heterogeneous $N_2O_5$ reactions on atmospheric aerosols at four Chinese sites: Improving model representation of uptake parameters"

_Atmospheric Chemistry and Physics, 2019_

## Referee Comment (RC1) · Anonymous Referee #1 · 2 Jan 2020

The authors present new measurements of the N2O5 uptake coefficient from field measurements in China. The additional measurements are of great value to the ongoing interpretation of N2O5 heterogeneous chemistry. The manuscript is well written and should be published following the authors attention to the following details:

Line 78: It would be helpful to clarify what "rural" refers to. If this is meant to denote a chemical regime, it would be helpful to classify by chemical composition (e.g., PM2.5, NOx, O3, CO, etc).

Lines 92 – 121: A reference to the original work of Bertram et al should be included here as the design and sampling approach appear to very closely replicate the tech-

nique described in Bertram et al., AMT 2009. Also, the uncertainty is a function of the surface area and RH, I find it nearly impossible that the uncertainty ranges only between 37-40% for the range of atmospheric conditions sampled. If this is correct, more discussion should be included.

Line 156: It is not clear how the uptake coefficient at 50M H2O which is essentially pure water is significantly larger than 0.03 (that measured for pure water in the laboratory). The authors should provide some discussion for how the 'measured' uptake coefficients are exceeding the rate for N2O5 with pure water? The limit observed in Hallquist, Thornton, ad Bertram and Thornton (all laboratory studies) correspond to reaching the upper limit of N2O5 uptake to pure water.

Line 158: Linear not liner.

Line 180: Correct use of the Bertram and Thornton parameterization involves calculating the aerosol water content for both the organic and inorganic components. Also, something doesn't seem to add up in the parameterization of the uptake coefficient. What value for V/S and KH were used and what temperature was this run at? It was my understanding that the parameterization could not exceed 0.03 based on the hydrolysis rate and henry's law coefficient used in the parameterization.

A note on empirical parameterizations: I think there is an opportunity to use field measurements to tune mechanistic parametrizations. Although, it seems unlikely that field measurements will do a better job than laboratory measurements at constraining rate coefficients or ratios of rate coefficients for the inorganic reactions. Laboratory experiments are designed to do this, using targeted simple 1-2 component systems and 1000x the aerosol surface area. I would expect the real power is looking at the variability in the hydrolysis rate or henry's law terms that are required to match the measurements with the models. This is where the complexity of the atmospheric aerosol will cause issues in laboratory parameterizations.

---

## Referee Comment (RC2) · Anonymous Referee #2 · 6 Feb 2020

Review of "Heterogeneous N2O5 reactions on atmospheric aerosols at four Chinese sites: Improving model representation of uptake parameters" This study developed an improved observation-based parameterization of N2O5 uptake coefficient and showed that the new parameterization improved the simulation results of NO2 and NO3- by the WRF-CMAQ model. The manuscript is generally well written. I think that it can be considered for publication after the authors address the following comments and suggestions.

1. Please clearly indicate the scope of application of the new parameterization. Is it applicable to China only or the whole world? I suggest that you apply the parameteri-

zation to all the sites shown in Fig. 3 to examine how it performs in other regions of the world. Even within China, please comment on whether the five sites used in this study are representative of China's general environmental conditions.

2. You only evaluated the CMAQ simulation results against $NO_2$ and $NO_3^-$ observations. Since you made many $N_2O_5$ and $ClNO_2$ measurements in this study, I strongly suggest that you also compare the simulation results with these data to better evaluate the performance of the new parameterization in CMAQ. In fact, I think the evaluation results of $N_2O_5$ may more directly reflect the performance of the $N_2O_5$ uptake parameterization.

3. Line 246-249: Your modeling domain covers the whole China and this sentence implies that you do have observational data in southern China. In this case, it looks strange that you only evaluated the simulation results over the North China Plain. I suggest that you provide a quantitative evaluation in southern China rather than just a speculation here.

4. Line 130-132: Although the detailed configuration of CMAQ has been described in a previous paper, I think it is still helpful to briefly describe some key configurations, especially those related to $NO_2$/$N_2O_5$/$ClNO_2$/$NO_3^-$ chemistry.

5. Fig. 2d: Obviously the curve does not fit the data points well. Could you justify why you select this formula?
* * *

---

## Author Comment (AC1) · 3 Mar 2020

**Response to Referee #1**

The authors present new measurements of the N2O5 uptake coefficient from field measurements in China. The additional measurements are of great value to the ongoing interpretation of N2O5 heterogeneous chemistry. The manuscript is well written and should be published following the authors attention to the following details:

Response: We appreciate the reviewer for the encouraging and helpful comments on our manuscript. We have made all of the suggested changes and clarifications. The reviewer's comments are in black, and our responses are in blue, and the changes in the manuscript are in red.

Line 78: It would be helpful to clarify what "rural" refers to. If this is meant to denote a chemical regime, it would be helpful to classify by chemical composition (e.g., PM2.5, NOx, O3, CO, etc).

Response: Heshan site was considered as a semi-rural site because the site is located outside towns and far from the urban area. The site was mostly affected by natural emission and some agriculture activities, with limited influences from the transport of anthropogenic emissions (e.g., industry and vehicles). The average concentrations of $PM_{2.5}$ and trace gases during the campaign were shown in Fig. 1b. We have revised the text to make it clearer, as follows,

"Field measurements of $\gamma_{N2O5}$ and related parameters were conducted at a semi-rural site (Heshan) in southern China from 22 February to 28 March 2017 and at a mountain site (Mt. Tai) in northern China from 11 March to 8 April 2018. Heshan site was located on a small hill (22.73°N, 112.92°E, 60 m a.s.l), surrounded by subtropical trees and some farmlands. A small city, Heshan, is 10 km to the northeast of the site, and three large cities, Guangzhou (the capital of Guangdong Province), Foshan and Jiangmen, are 80 km to the northeast, 50 km to the northeast and 30 km to the southwest of the site, respectively. The site is affected by vehicle emissions from three highways and two provincial roads within 10 km and some residential/agriculture activities in the area, and thus was considered as a semi-rural site."

"All the sites are regionally representative sites, as they are situated in an area with limited anthropogenic influences (Tham et al., 2016; Wang et al., 2017a; Yun et al., 2018; Wang et al., 2016). The detailed information of the sampling sites, instrumentation, and $\gamma_{N2O5}$ determination approach have been described in the previous publications (Wang et al., 2016; Tham et al., 2016; Wang et al., 2017a), and site descriptions are briefly summarized in the SI. The locations of all the measurement sites are shown on the map in Fig. 1a. The statistics of the trace gases and $PM_{2.5}$ measured during the campaigns were summarized in Fig. 1b, representing general pollution conditions at these sites. The mean concentration of $O_3$, NOx and $PM_{2.5}$ at these sites ranged from 43 to 80 ppbv, 2.4 to 14.5 ppbv and 9.9 to 80.2 µg m$^{-3}$, respectively."

Lines 92 – 121: A reference to the original work of Bertram et al should be included here as the design and sampling approach appear to very closely replicate the technique described in

Bertram et al., AMT 2009. Also, the uncertainty is a function of the surface area and RH, I find it nearly impossible that the uncertainty ranges only between 37-40% for the range of atmospheric conditions sampled. If this is correct, more discussion should be included.

Response: Thanks for the suggestions. The reference to Bertram et al. has been added to the manuscript. The uncertainty was described in detail in our previous paper (Wang et al., 2018) and $\gamma_{N2O5}$ was found to be most sensitive to RH, and more discussion was added in the revised manuscript.

The revised text reads,

"The uptake coefficient of $N_2O_5$, $\gamma_{N2O5}$, was derived from the direct measurement of the loss rate coefficient of $N_2O_5$ on ambient aerosols using an aerosol flow tube based on the design of Bertram et al. (2009), with some improvements and coupling with an iterative box model for polluted environments (Wang et al., 2018)."

"The uncertainty introduced by $S_a$ measurement would be propagated to an uncertainty of 30% in the calculated $\gamma_{N2O5}$. The improvement with the use of the box model in the system could minimize the influences from the variability of ambient conditions as well as fresh NO emission, and a Monte Carlo approach was employed to evaluate the uncertainty from different parameters (Wang et al., 2018). The estimated uncertainty ranged from 37% to 40% at $\gamma_{N2O5}$ around 0.03 and from 34% to 65% at $\gamma_{N2O5}$ around 0.01 with $S_a = 1000$ $\mu m^2$ $cm^{-3}$ when RH varied from 20% to 70 %, and could be higher at RH>70%."

References
*Bertram, T., Thornton, J., and Riedel, T.: An experimental technique for the direct measurement of $N_2O_5$ reactivity on ambient particles, Atmospheric Measurement Techniques, 2, 231-242, 2009a.*
*Wang, W., Wang, Z., Yu, C., Xia, M., Peng, X., Zhou, Y., Yue, D., Ou, Y., and Wang, T.: An in situ flow tube system for direct measurement of $N_2O_5$ heterogeneous uptake coefficients in polluted environments, Atmospheric Measurement Techniques, 11, 5643-5655, 10.5194/amt-11-5643-2018, 2018.*

Line 156: It is not clear how the uptake coefficient at 50M H2O which is essentially pure water is significantly larger than 0.03 (that measured for pure water in the laboratory). The authors should provide some discussion for how the 'measured' uptake coefficients are exceeding the rate for N2O5 with pure water? The limit observed in Hallquist, Thornton, ad Bertram and Thornton (all laboratory studies) correspond to reaching the upper limit of N2O5 uptake to pure water.

Response: Thanks for the suggestions. We were aware that many previous laboratory studies (Hallquist et al., 2000; Hallquist et al., 2003; Thornton et al., 2003; Thornton et al., 2005) had observed the $N_2O_5$ uptake on different particles (e.g., sulfuric acid, ammonium sulfate/bisulfate, malonic acid and sea salt) showing strong dependence on RH below 50% but reaching an upper

limit around 0.036 at RH around 50%-80%. However, there were also higher $\gamma_{N2O5}$ values measured by different laboratory studies. For example, Mozurkewich and Calvert (1988) reported an upper limit of $\gamma_{N2O5}$ on $NH_4HSO_4$ particles of 0.056 at RH = 55% at 293K, which increased to around 0.1 at 274K. Kane et al. (2001) observed a strong RH dependent $\gamma_{N2O5}$ on $NH_4HSO_4$ particles, increasing from 0.018 to 0.069 with RH from 56% to 99%. Moreover, several field measurements also reported the $\gamma_{N2O5}$ exceeding 0.04 at high RH or water molarity (e.g. Philips et al., 2016; McDuffie et. al, 2018; Wang H et al., 2017; Tham et. al, 2018), and some of them found the similar positive relationship between $\gamma_{N2O5}$ and water molarity (McDuffie et. al, 2018). Although uncertainties may exist in the calculation of aerosol surface and uptake coefficient at high ambient RH conditions, our results here indicate that the aerosol water content strongly affects the activity of $N_2O_5$ uptake, and the $N_2O_5$ hydrolysis is always limited by aerosol water content. It is unclear what exact mechanism or process (e.g., phase change different from laboratory-made particles, or acidity involved) promote more effective uptake on ambient aerosols at higher RH and water content condition due to the limited measurement data in laboratory and field. More detailed investigation of $N_2O_5$ uptake on nano-size water/aerosol droplets in the real (or close to real) ambient conditions are clearly warranted.

More discussion is added in the revised text, as follows:

"Although the positive correlation of $\gamma_{N2O5}$ with the humidity or aerosol water has been observed in the low range in previous laboratory studies, the $\gamma_{N2O5}$ reached plateaus at a value around 0.036 at RH> 50% or $[H_2O] > 15$ M (Hallquist et al., 2003; Thornton et al., 2003; Bertram & Thornton, 2009). In contrast, other laboratory studies also measured higher $\gamma_{N2O5}$ values on $NH_4HSO_4$ particles. For example, Mozurkewich and Calvert (1988) reported an upper limit of $\gamma_{N2O5}$ of 0.056 at RH = 55% at 293 K, which increased to around 0.1 at 274 K. Kane et al. (2001) observed a strong RH dependent $\gamma_{N2O5}$, increasing from 0.018 to 0.069 with RH from 56% to 99%, which is largely consistent with the field results in the present study. Moreover, several field measurements also observed $\gamma_{N2O5}$ value exceeding 0.04 at high RH or water molarity (e.g. Philips et al., 2016; McDuffie et. al, 2018a; Wang et al., 2017c; Tham et. al, 2018), and some of them also found the similar positive relationship between $\gamma_{N2O5}$ and water molarity (McDuffie et. al, 2018a). Although uncertainties may exist in the calculation of aerosol surface and uptake coefficient at high ambient RH conditions, our results with a consistently increasing trend of $\gamma_{N2O5}$ with $[H_2O]$ from below 10 M up to 50 M suggest that the aerosol water content strongly affects the activity of $N_2O_5$ uptake, and that $N_2O_5$ hydrolysis is always limited by aerosol water content under all the encountered ambient conditions. Since limited measurement data of $\gamma_{N2O5}$ from laboratory and fields are available at RH > 80% condition, it is unclear what exact mechanism or process (e.g., phase change different from laboratory-made particles or acidity involved) promote more effective uptake on ambient aerosols at higher aerosol water content condition. Therefore, more detailed investigations of $N_2O_5$ uptake on nano-size water/aerosol droplets in the real (or close to real) ambient conditions are clearly warranted."

References
*Kane, S. M., Caloz, F., and Leu, M.-T.: Heterogeneous uptake of gaseous $N_2O_5$ by $(NH_4)_2SO_4$, $NH_4HSO_4$, and $H_2SO_4$ aerosols, The Journal of Physical Chemistry A, 105, 6465-6470, 2001.*

*Hallquist, M., Stewart, D. J., Baker, J., and Cox, R. A.: Hydrolysis of N2O5 on submicron sulfuric acid aerosols, The Journal of Physical Chemistry A, 104, 3984-3990, 2000.*

*Hallquist, M., Stewart, D. J., Stephenson, S. K., and Anthony Cox, R.: Hydrolysis of N2O5 on sub-micron sulfate aerosols, Physical Chemistry Chemical Physics, 5, 3453, 10.1039/b301827j, 2003.*

*Mozurkewich, M., and Calvert, J. G.: Reaction probability of $N_2O_5$ on aqueous aerosols, Journal of Geophysical Research: Atmospheres, 93, 15889-15896, 1988.*

*McDuffie, E. E., Fibiger, D. L., Dubé, W. P., Lopez‐Hilfiker, F., Lee, B. H., Thornton, J. A., Shah, V., Jaeglé, L., Guo, H., and Weber, R. J.: Heterogeneous $N_2O_5$ uptake during winter: Aircraft measurements during the 2015 WINTER campaign and critical evaluation of current parameterizations, Journal of Geophysical Research: Atmospheres, 123, 4345-4372, 2018.*

*Phillips, G. J., Thieser, J., Tang, M., Sobanski, N., Schuster, G., Fachinger, J., Drewnick, F., Borrmann, S., Bingemer, H., and Lelieveld, J.: Estimating $N_2O_5$ uptake coefficients using ambient measurements of $NO_3$, $N_2O_5$, $ClNO_2$ and particle-phase nitrate, Atmospheric Chemistry and Physics, 16, 13231-13249, 2016.*

*Tham, Y. J., Wang, Z., Li, Q., Wang, W., Wang, X., Lu, K., Ma, N., Yan, C., Kecorius, S., and Wiedensohler, A.: Heterogeneous $N_2O_5$ uptake coefficient and production yield of $ClNO_2$ in polluted northern China: roles of aerosol water content and chemical composition, Atmospheric Chemistry and Physics, 18, 13155-13171, 2018.*

*Thornton, J. A., Braban, C. F., and Abbatt, J. P. D.: $N_2O_5$ hydrolysis on sub-micron organic aerosols: the effect of relative humidity, particle phase, and particle size, Physical Chemistry Chemical Physics, 5, 4593-4603, 10.1039/B307498F, 2003.*

*Thornton, J. A., and Abbatt, J. P. D.: N2O5 Reaction on Submicron Sea Salt Aerosol: Kinetics, Products, and the Effect of Surface Active Organics, The Journal of Physical Chemistry A, 109, 10004-10012, 10.1021/jp054183t, 2005.*

*Wang, H., Lu, K., Chen, X., Zhu, Q., Chen, Q., Guo, S., Jiang, M., Li, X., Shang, D., and Tan, Z.: High $N_2O_5$ concentrations observed in urban Beijing: Implications of a large nitrate formation pathway, Environmental Science & Technology Letters, 4, 416-420, 2017c.*

Line 158: Linear not liner.

Response: Corrected.

Line 180: Correct use of the Bertram and Thornton parameterization involves calculating the aerosol water content for both the organic and inorganic components. Also, something doesn't seem to add up in the parameterization of the uptake coefficient. What value for V/S and KH were used and what temperature was this run at? It was my understanding that the parameterization could not exceed 0.03 based on the hydrolysis rate and henry's law coefficient used in the parameterization.

A note on empirical parameterizations: I think there is an opportunity to use field measurements to tune mechanistic parametrizations. Although, it seems unlikely that field measurements will do a better job than laboratory measurements at constraining rate coefficients or ratios of rate coefficients for the inorganic reactions. Laboratory experiments are designed to do this, using targeted simple 1-2 component systems and 1000x the aerosol surface area. I would expect the

real power is looking at the variability in the hydrolysis rate or henry's law terms that are required to match the measurements with the models. This is where the complexity of the atmospheric aerosol will cause issues in laboratory parameterizations.

Response: Thanks for the helpful comments. First, we want to clarify that the $V_a/S_a$ values varied from $3.30\times10^{-8}$ m to $9.29\times10^{-8}$ m in different cases, and were measured from the particle number and size distribution by WPS or SMPS in different campaigns. The Henry's law dimensionless constant of $K_H$ was taken as 51, following the Bertram and Thornton parameterization. The temperature ranged from 3℃ to 28℃ in different cases in the five campaigns.

Secondly, as shown in the Bertram and Thornton parameterization, $\gamma_{N2O5}$ was linearly dependent on the $V_a/S_a$ value by assuming a volume-limited reaction. Because the average $V_a/S_a$ of $3.75\times10^{-8}$ m was used in their study, the upper limit of $\gamma_{N2O5}$ at high $[H_2O]$ from the parameterization was around 0.036. And the upper limit of $\gamma_{N2O5}$ would increase proportionally to the increase of V/S value.

In addition, since Bertram and Thornton observed that the $\gamma_{N2O5}$ was insensitive to RH above 50%, they treated the reaction rate coefficient of $[N_2O_5]$ with $[H_2O]$ as an inverse exponential function of $[H_2O]$. In the present study, we use the original second-order rate definition of water reaction term as $k_{2f}[H_2O]$, which leads to a linear dependence of $\gamma_{N2O5}$ on $[H_2O]$ and is consistent with our field observation results. The secondary order rate constant $k_{2f}$ was fitted to be $(3.0\pm0.4)\times10^4$ $M^{-1}$ $s^{-1}$, which is in reasonable agreement with the values of $(2.7-3.8)\times10^4$, $\sim3.9\times10^4$ and $2.6\times10^4$ $M^{-1}$ $s^{-1}$ determined from ammonium bisulfate, ammonium sulfate (Gaston et al., 2016) and aqueous organic acid particles (Thornton et al. 2003), respectively.

We agree with the reviewer that it would be an opportunity to use the field data to tune mechanistic parameters such as Henry's law term. Different $K_H$ values have been used in previous studies, e.g., ~50 (e.g., Hallquist et al., 2003; Bertram and Thornton, 2009) or ~120 (corresponding to a Henry's law of 5M/atm at 298K) (e.g., Gaston et al., 2014; 2016; Griffiths et al., 2009). The $\gamma_{N2O5}$ in the parameterization is also linearly dependent on the $K_H$, and thus an increase of $K_H$ value would proportionally increase the $\gamma_{N2O5}$ value but cannot account for the large variability of measured $\gamma_{N2O5}$ comparing to the parameterized values. In addition, there is lack of an explicit function of effective Henry's law constant for $N_2O_5$ to include the 'salting-in' effect and other processes, therefore in the present study, we use the value of 51 suggested by Bertram and Thornton and enclosed the influences from the aerosol composition in the last 'chemical' term. The derived empirical ratios in the last 'chemical' term not only represent the competing ratio of these reactions but also include other unspecified effects or influences (e.g., organic coating, mixing state, other nucleophiles reactions, etc.).

We have clarified this and added more discussion in the revised text, as follows,

"where $V_a/S_a$ is the measured aerosol volume to surface area ratio, ranging from $3.30\times10^{-8}$ to $9.29\times10^{-8}$ m in the five campaigns; $K_H$ is Henry's law coefficient, taken as 51 (Bertram &

Thornton, 2009; Fried et al., 1994);"

"In view of the linear dependence of $\gamma_{N2O5}$ on the aerosol water content in this study and reaction mechanism (Bertram & Thornton, 2009), the second-order reaction rate coefficient with water (refer to k'$_{2f}$ in Eq. (2) and Eq. (3)) was fitted as a linear function of [H$_2$O], as $(3.0\pm0.4) \times 10^4 \times$ [H$_2$O]. This value is in reasonable agreement with the values of $(2.7{-}3.8) \times 10^4$, $\sim 3.9 \times 10^4$, and $2.6 \times 10^4$ M$^{-1}$ s$^{-1}$ determined from ammonium bisulfate, ammonium sulfate (Gaston et al., 2016) and aqueous organic acid particles (Thornton et al. 2003), respectively. Compared to original BT09 (Eq. (3)), the newly fitted k'$_{2f}$ is smaller for [H$_2$O] < 38 M, but become higher with the increasing of aerosol water content (Fig. S2). Different dimensionless K$_H$ values have been used in previous studies, e.g., ~50 (e.g., Hallquist et al., 2003; Bertram and Thornton, 2009) or ~120 (e.g., Gaston et al., 2014; 2016; Griffiths et al., 2009), which correspond to a Henry's law constant of 2 or 5 M atm$^{-1}$ at 298K. As $\gamma_{N2O5}$ in the parameterization is linearly dependent on the K$_H$, an increase of K$_H$ value would proportionally increase the $\gamma_{N2O5}$ value but cannot account for the large variability of measured $\gamma_{N2O5}$ values. Given the lack of an explicit function of effective Henry's law constant for N$_2$O$_5$ to include the different process (e.g., 'salting-in' effect and surface processes), we use the value of 51 suggested by Bertram and Thornton (2009) and enclose those effects from the aerosol composition in the last 'chemical' term. The derived empirical ratios in the last 'chemical' term not only represent the competing ratio of these reactions but also include other unspecified effects or processes (e.g., organic coating, mixing state, other nucleophiles reactions, etc.).

References
Bertram, T., and Thornton, J.: Toward a general parameterization of N$_2$O$_5$ reactivity on aqueous particles: the competing effects of particle liquid water, nitrate and chloride, Atmospheric Chemistry and Physics, 9, 8351-8363, 2009.
Fried, A., Henry, B. E., Calvert, J. G., and Mozurkewich, M.: The reaction probability of N$_2$O$_5$ with sulfuric-acid aerosols at stratospheric temperatures and compositions, J. Geophys. Res., 99, 3517–3532, 1994.
Gaston, C. J., Thornton, J. A., and Ng, N. L.: Reactive uptake of N$_2$O$_5$ to internally mixed inorganic and organic particles: the role of organic carbon oxidation state and inferred organic phase separations, Atmos. Chem. Phys., 14, 5693–5707, 2014.
Gaston, C. J., and Thornton, J. A.: Reacto-diffusive length of N$_2$O$_5$ in aqueous sulfate-and chloride-containing aerosol particles, The Journal of Physical Chemistry A, 120, 1039-1045, 2016.
Griffiths, P. T., Badger, C. L., Cox, R. A., Folkers, M., Henk, H. H., and Mentel, T. F.: Reactive uptake of N$_2$O$_5$ by aerosols containing dicarboxylic acids. Effect of particle phase, composition, and nitrate content, The Journal of Physical Chemistry A, 113, 5082-5090, 2009.
Hallquist, M., Stewart, D. J., Stephenson, S. K., and Anthony Cox, R.: Hydrolysis of N$_2$O$_5$ on sub-micron sulfate aerosols, Physical Chemistry Chemical Physics, 5, 3453, 10.1039/b301827j, 2003.
Thornton, J. A., Braban, C. F., and Abbatt, J. P. D.: N$_2$O$_5$ hydrolysis on sub-micron organic aerosols: the effect of relative humidity, particle phase, and particle size, Physical Chemistry Chemical Physics, 5, 4593-4603, 10.1039/B307498F, 2003.

---

## Author Comment (AC2) · 3 Mar 2020

**Response to Anonymous Referee #2**

This study developed an improved observation-based parameterization of N2O5 uptake coefficient and showed that the new parameterization improved the simulation results of NO2 and NO3- by the WRF-CMAQ model. The manuscript is generally well written. I think that it can be considered for publication after the authors address the following comments and suggestions.

Response: We appreciate the reviewer for the helpful comments on our manuscript. We have made all of the suggested changes and clarifications. The reviewer's comments are in black and our responses are in blue, and the changes in the manuscript are in red.

1. Please clearly indicate the scope of application of the new parameterization. Is it applicable to China only or the whole world? I suggest that you apply the parameterization to all the sites shown in Fig. 3 to examine how it performs in other regions of the world. Even within China, please comment on whether the five sites used in this study are representative of China's general environmental conditions.

Response: We think the empirical parameterization should be applicable to different areas in China, especially those polluted regions. The four sites were all located in semi-rural areas with regional representativeness in north or south China. For the sites other than described in this study in Fig. 3, the detailed experimental data such as inorganic compositions and $V_a/S_a$ of each data points are not available in the literature, and thus it is not possible for us to evaluate and compare the new empirical parameters at all other sites in the world. We advocate further validation of the parameterization derived from the present study in other regions of the world.

We added more information to clarify it, as follows,

"All the sites are regionally representative sites, as they are situated in an area with limited anthropogenic influences (Tham et al., 2016; Wang et al., 2017a; Yun et al., 2018; Wang et al., 2016). The detailed information of the sampling sites, instrumentation and $\gamma_{N2O5}$ determination approach have been described in the previous publications (Wang et al., 2016; Tham et al., 2016; Wang et al., 2017a), and site descriptions are briefly summarized in the SI. The locations of all the measurement sites are shown on the map in Fig. 1a. The statistics of the trace gases and $PM_{2.5}$ measured during the campaigns were summarized in Fig. 1b, representing general pollution conditions at these sites. The mean concentration of $O_3$, NOx and $PM_{2.5}$ at these sites ranged from 43 to 80 ppbv, 2.4 to 14.5 ppbv and 9.9 to 80.2 $\mu g\ m^{-3}$, respectively."

"More tests of this empirical parameterization are warranted for other locations/seasons in China and other parts of the world."

References

*Tham, Y. J., Wang, Z., Li, Q., Yun, H., Wang, W., Wang, X., Xue, L., Lu, K., Ma, N., and Bohn, B.: Significant concentrations of nitryl chloride sustained in the morning: investigations of*

*the causes and impacts on ozone production in a polluted region of northern China, Atmospheric chemistry and physics, 2016.*

*Wang, T., Tham, Y. J., Xue, L., Li, Q., Zha, Q., Wang, Z., Poon, S. C., Dubé, W. P., Blake, D. R., and Louie, P. K.: Observations of nitryl chloride and modeling its source and effect on ozone in the planetary boundary layer of southern China, Journal of Geophysical Research: Atmospheres, 121, 2476-2489, 2016.*

*Wang, Z., Wang, W., Tham, Y. J., Li, Q., Wang, H., Wen, L., Wang, X., and Wang, T.: Fast heterogeneous $N_2O_5$ uptake and $ClNO_2$ production in power plant and industrial plumes observed in the nocturnal residual layer over the North China Plain, Atmospheric Chemistry and Physics, 17, 12361-12378, 10.5194/acp-17-12361-2017, 2017a.*

*Yun, H., Wang, W., Wang, T., Xia, M., Yu, C., Wang, Z., Poon, S. C. N., Yue, D., and Zhou, Y.: Nitrate formation from heterogeneous uptake of dinitrogen pentoxide during a severe winter haze in southern China, Atmospheric Chemistry and Physics, 18, 17515-17527, 10.5194/acp-18-17515-2018, 2018.*

2. You only evaluated the CMAQ simulation results against NO2 and NO3- observations. Since you made many N2O5 and ClNO2 measurements in this study, I strongly suggest that you also compare the simulation results with these data to better evaluate the performance of the new parameterization in CMAQ. In fact, I think the evaluation results of N2O5 may more directly reflect the performance of the N2O5 uptake parameterization.

Response: We agree with the reviewer that the comparison of simulated $N_2O_5$ could provide a more direct evaluation of the performance of the new parameterization. Because of the short lifetime of $N_2O_5$ (usually several to ten minutes) (Tham et al., 2018; Yun et al., 2019), it is more prone to be affected by local emissions and fluctuation of meteorological parameters. For the regional models with a grid resolution of tens of kilometers, it is difficult for the regional model to capture the variation of $N_2O_5$. We made a comparison of the statistic results of simulated $N_2O_5$ concentrations in winter 2017 with those observed in the wintertime at various locations of China, including two in the North China Plain (Beijing and Wangdu in Hebei province) and two in southern China (Tai Mao Shan and Heshan). With the new parameterization, the WRF-Chem model can better simulate the average concentrations and variation ranges of $N_2O_5$ at these locations. The results were added as the new Figure 6, and the discussions were also added, as follows,

"In addition to $NO_2$ and $NO_3^-$, we also compared the simulated $N_2O_5$ concentrations for December 2017 with those observed in the wintertime at various locations of China, including two in the North China Plain (Beijing and Wangdu in Herbei province) and two in southern China (Tai Mao Shan and Heshan). As shown in Figure 6, with the new parameterization, the WRF-CMAQ model can better simulate the average concentration and variation range of $N_2O_5$ at these locations. Overall, the new parameterization has significantly reduced the discrepancies between the modelled and observed concentrations of $NO_2$, $N_2O_5$ and $NO_3^-$ at our study sites and periods in both northern and southern China. More tests of this empirical parameterization are warranted for other locations/seasons in China and other parts of the world."

[Figure]

**Figure 6. Comparison of the simulated N$_2$O$_5$ concentrations by the CMAQ model for December 2017 with the wintertime observation results from four sites in China. The field observations were conducted in December 2017 at Wangdu, January 2018 at Beijing, January 2018 at Heshan and November 2013 at Tai Mo Shan. The columns and error bars represent the average value and standard deviation, respectively.**

3. Line 246-249: Your modeling domain covers the whole China and this sentence implies that you do have observational data in southern China. In this case, it looks strange that you only evaluated the simulation results over the North China Plain. I suggest that you provide a quantitative evaluation in southern China rather than just a speculation here.

Response: We focused on northern China for Dec 2017 in evaluating the new parameterization in part because of the availability of the unique regional observations of PM$_{2.5}$ nitrate aerosol. We agree that comparisons with N$_2$O$_5$ observations are valuable. As responded to comment #2, we have further compared the model simulation in southern China with our previous field observations at Heshan and Tai Mo Shan, and the comparison results of N$_2$O$_5$ are now included in the revised text, see the response to above comment #2.

4. Line 130-132: Although the detailed configuration of CMAQ has been described in a previous paper, I think it is still helpful to briefly describe some key configurations, especially those related to NO2/N2O5/ClNO2/NO3- chemistry.

Response: Adopted and a brief description of CMAQ configuration is added.

"In addition, the Community Multiscale Air Quality (CMAQ) model (v5.1) was employed to evaluate the uptake parameterization. Two simulations (default and revised) were conducted. In the default case, the N$_2$O$_5$ uptake and ClNO$_2$ production were calculated based on the parameterization of Bertram and Thornton (2009). In the revised case, the new parameterization derived in this study was used. Other model configurations were the same. The SAPRC07tic

gas mechanism and AERO6i aerosol mechanism was used. Weather Research and Forecasting (WRF) (v4.0) was applied to generate the meteorological inputs for the CMAQ simulations. The anthropogenic emission inputs were generated based on the local Chinese emission inventory (Zhao et al. 2018) and the INTEX-B dataset for Asia (Zhang et al., 2009). The high-resolution chloride emission inventory for China from Fu et al. (2018) was also included. More details for model configuration can be found in Fu et al. (2019). The simulation domain covers China with a resolution of 36×36 km (Fig. S1), based on a Lambert projection with two true latitudes of 25°N and 40°N."

References

Zhao, B., Zheng, H., Wang, S., Smith, K. R., Lu, X., Aunan, K., Gu, Y., Wang, Y., Ding, D., and Xing, J.: Change in household fuels dominates the decrease in $PM_{2.5}$ exposure and premature mortality in China in 2005–2015, Proceedings of the National Academy of Sciences, 115, 12401-12406, 2018.

Zhang, Q., Streets, D. G., Carmichael, G. R., He, K., Huo, H., Kannari, A., Klimont, Z., Park, I., Reddy, S., and Fu, J.: Asian emissions in 2006 for the NASA INTEX-B mission, Atmospheric Chemistry and Physics, 9, 5131-5153, 2009.

Fu, X., Wang, T., Wang, S., Zhang, L., Cai, S., Xing, J., and Hao, J.: Anthropogenic emissions of hydrogen chloride and fine particulate chloride in China, Environmental science & technology, 52, 1644-1654, 2018.

Fu, X., Wang, T., Zhang, L., Li, Q., Wang, Z., Xia, M., Yun, H., Wang, W., Yu, C., and Yue, D.: The significant contribution of HONO to secondary pollutants during a severe winter pollution event in southern China, Atmospheric Chemistry and Physics, 19, 1-14, 2019.

5. Fig. 2d: Obviously the curve does not fit the data points well. Could you justify why you select this formula?

Response: Thanks for pointing this out. The curve was fitted just to check whether the trend following the relationship between $\gamma_{N2O5}$ and $Cl^-/NO_3^-$ derived from laboratory studies. The discrepancy of the data and the curve shows the $Cl^-$ enhancement in our study is not as strong or obvious as that found in other laboratory studies. To avoid misleading the reader, the fitting curve is removed from Fig. 2d.

---

## Author Response (AR2)

**Response to Editor**

Regarding to the uncertainty introduced by Sa measurement, I agree with the reviewer that it is nearly impossible to be only between 37-40% if you include the contributions of all the factors. For instance, the ambient aerosol surface area density was calculated from the dry particle size distributions. For typical electrical mobility based aerosol size distribution measurement, the charging fractions in atmospheric conditions can be different from instrument-default conditions and then leads to significant uncertainty. There has been quite some discussions on this topic. In addition, the unknown morphology of aerosol particles can introduce significant uncertainty when deriving aerosol surface area from particle number based measurements.

In addition to just state that "the uncertainty introduced by Sa measurement would be propagated to an uncertainty of 30%......to evaluate the uncertainty from different parameters (Wang et al., 2018)", therefore, I suggest that you clearly and concisely specify which factors were included in your uncertainty evaluation. This will help the readers to understand the reported uncertainty range.

**Response**: We appreciate the editor for helpful suggestions on the uncertainty estimation. We agree with the editor that there are significant uncertainties in the measurement of particle number size distribution (PNSD), resulting from the charging fraction, mobility particle classification accuracy, particle counting efficiency, and sampling flow variability, etc. (*Jiang et al., 2014; Kuang et al., 2016; Widensohler et al. 2012*). As suggested by Kuang et al (*2016*), the uncertainty of typical PNSD measurement would be dominated primarily by aerosol flow rate and charging efficiency uncertainties, and the size-dependent aerosol charging efficiency is typically characterized by an accuracy of $\pm 10\%$ (*Jiang et al. 2014*). The sizing accuracy of the instruments and flow rate variability was usually within $\pm 2$ %. Besides, Widensohler et al. (*2012*) have reported that the PNSD from 20 to 200 nm determined by mobility particle size spectrometers of different designs were within an uncertainty range of around $\pm 10\%$ under the controlled conditions. Therefore, it is reasonable to assume a 20% uncertainy in PNSD for the size range that dominated the particle surface area. By assuming a conservative uncertainty of 15% for the hygroscopic growth at RH<90% (*Liu et al., 2014*), the uncertainty associated with the aerosol surface area determination was estimated to be approximately 30%. However, the uncertainty introduced by the unknown morphology of particles is difficult to be quantified and is not accounted in the Sa determination, therefore the uncertainty reported here can be considered as a lower limit. We add the caveat in the revised text.

The uncertainty evaluation in the $N_2O_5$ uptake determination from the flow tube system in Wang et al. (*2018*) considered the uncertainties in mean residence time, wall loss variability with ambient RH, input $N_2O_5$ concentration, and variability of ambient conditions of NO, $NO_2$, $O_3$, and VOCs during a measurement cycle. The estimated uncertainty with a fixed Sa of 1000 $\mu m^2$ $cm^{-3}$ ranged from 21% to 27% at $\gamma_{N2O5}$ around 0.03 and from 17% to 58% at $\gamma_{N2O5}$ around 0.01 when RH varied from 20% to 70%. Therefore, the overall propagated uncertainty was in the range of 37% to 40% and 34% to 65% at $\gamma_{N2O5}$ around 0.03 and 0.01, respectively. For RH>70%, the uncertainty in the $\gamma_{N2O5}$ determination could be higher, and it may be up to 100% for RH around 90% (*Wang et al., 2017*). We have revised the text to clarify these factors considered in the uncertainty evaluation.

The revised text reads:

[revised manuscript text omitted]